# EQUIVARIANT GRAPH NETWORK APPROXIMATIONS OF HIGH-DEGREE POLYNOMIALS FOR FORCE FIELD PREDICTION

## ABSTRACT

Equivariant deep models have recently been employed to predict atomic potentials and force fields in molecular dynamics. A key advantage of these models is their ability to learn from data without requiring explicit physical modeling. Nevertheless, use of models obeying underlying physics can not only lead to better performance, but also yield physically interpretable results. In this work, we propose a new equivariant network, known as PACE, to incorporate many-body interactions by making use of the Atomic Cluster Expansion (ACE) mechanism. To provide a solid foundation for our work, we perform theoretical analysis showing that our proposed message passing scheme can approximate any equivariant polynomial functions with constrained degree. By relying physical insights and theoretical foundations, we show that our model achieves state-of-the-art performance on atomic potential and force field prediction tasks on commonly used benchmarks.

## 1 INTRODUCTION

Deep learning has led to notable progress in computational quantum chemistry tasks, such as predicting atomic potentials and force fields in molecular systems (Zhang et al., 2023). Fast and accurate prediction of energy and force is desired, as it plays crucial roles in advanced applications such as material design and drug discovery. However, it is insufficient to rely solely on learning from data, as there are physics challenges that must be taken into consideration. For example, to better consider symmetries inherent in 3D molecular structures, equivariant graph neural networks (GNNs) have been developed in recent years. By using equivariant features and equivariant operations, SE(3)-equivariant GNNs ensure equivariance of permutation, translation and rotation. Thus, their internal features and predictions transform accordingly as the molecule is rotated or translated. Existing equivariant GNNs can specialize in handling features with either rotation order $\ell = 1$ (Schütt et al., 2021; Jing et al., 2021; Satorras et al., 2021; Du et al., 2022; 2023; Thölke and Fabritiis, 2022) or higher rotation order $\ell > 1$ (Thomas et al., 2018; Fuchs et al., 2020; Liao and Smidt, 2023; Batzner et al., 2022; Batatia et al., 2022a;b; Yu et al., 2023b; Unke et al., 2021; Yu et al., 2023a), while invariant methods only consider rotation order $\ell = 0$ (Schütt et al., 2017; Smith et al., 2017; Chmiela et al., 2017; Zhang et al., 2018a;b; Schütt et al., 2018; Ying et al., 2021; Luo et al., 2023; Gasteiger et al., 2020; Liu et al., 2022; Gasteiger et al., 2021; Wang et al., 2022; Lin et al., 2023; Yan et al., 2022; Liu et al., 2021). Generally, methods with higher rotation order exhibit improved performance but at the cost of higher computational complexity. In addition to rotation order, some equivariant methods (Musaelian et al., 2023; Batatia et al., 2022a;b) also consider many-body interactions in their model design. These approaches follow traditional principles (Brown et al., 2004; Braams and Bowman, 2009) of decomposing the potential energy surface (PES) as a linear combination of body-ordered functions. In contrast to standard message passing (Gilmer et al., 2017) that considers interactions between two atoms in each message, many-body methods aim to incorporate the interactions of multiple atoms surrounding the central node.

In this work, we present a novel equivariant network that incorporates many-body interactions based on the Atomic Cluster Expansion (ACE) mechanism (Drautz, 2019; Dusson et al., 2022; Kovács et al., 2021). We conduct a theoretical analysis, demonstrating the capability of our proposed message passing scheme to effectively approximate equivariant polynomial functions within a constrained degree, thereby establishing a solid foundation for our model. Our method is termed

PACE as it is based on polynomial function approximation and ACE. To evaluate the performance and generalization capabilities of our approach, we assess our model on two molecular dynamics simulation datasets, namely rMD17 and 3BPA, and obtain consistent performance enhancements. Notably, we achieve state-of-the-art performance in energy prediction across all molecules and achieve superior force prediction accuracy in 50% of the molecules, affirming the significance and potential of our proposed approach.

## 2 BACKGROUND AND RELATED WORK

### 2.1 SYMMETRIES AND EQUIVARIANCE

Considering physical symmetries in machine learning models is crucial for solving quantum chemistry problems, as various quantum properties of molecules exhibit inherent equivariance or invariance to symmetry transformations. For example, if we rotate a molecule in 3D space, forces acting on atoms rotate accordingly while the total energy of the molecule remains invariant. According to group theory (Bronstein et al., 2021) which analyzes symmetries of geometry and physics, given a group $G$ and group action $*$, we say $f : Q \rightarrow Y$ is $G$-equivariant if $f(g * \mathbf{q}) = g * f(\mathbf{q})$ for any $\mathbf{q} \in Q, g \in G$. If $f(g * \mathbf{q}) = f(\mathbf{q})$ holds, we say $f$ is $G$-invariant.

To encode geometric information of molecules into SE(3)-equivariant features, spherical harmonics are used for its equivariance property. Specifically, we use real spherical harmonic basis functions $Y$ to encode an orientation $\hat{r}_{ij}$ between a node pair. If the molecule is rotated by a rotation matrix $R$ in 3D coordinate system, then we have:

$$\mathbf{q}^\ell Y^\ell(R\hat{r}_{ij}) = (\boldsymbol{D}^\ell(R)\mathbf{q}^\ell)Y^\ell(\hat{r}_{ij}), \tag{1}$$

where $\ell \in [0, L]$ is the degree, $\mathbf{q}^\ell$ with size $2\ell+1$ denotes coefficients of spherical harmonics, and the Wigner D-matrix $\boldsymbol{D}^\ell(R)$ with size $(2\ell + 1) \times (2\ell + 1)$ specifies the corresponding rotation acting on coefficients $\mathbf{q}^\ell$. In practice, equivariant features are often represented by irreducible representations (irreps), which correspond to the coefficients of real spherical harmonics.

### 2.2 EQUIVARIANT GRAPH NEURAL NETWORKS

In recent years, equivariant graph neural networks have been developed for 3D molecular representation learning, as they are capable of effectively incorporating the symmetries required by the specific task. Existing equivariant 3D GNNs can be broadly classified into two categories, depending on whether they utilize order $\ell = 1$ equivariant features or higher order $\ell > 1$ equivariant features. Methods belonging to the first category (Satorras et al., 2021; Schütt et al., 2021; Deng et al., 2021; Jing et al., 2021; Thölke and Fabritiis, 2022) achieve equivariance by applying constrained operations on order 1 vectors, such as vector scaling, summation, linear transformation, vector product, and scalar product.

The second category of methods (Thomas et al., 2018; Fuchs et al., 2020; Liao and Smidt, 2023; Batzner et al., 2022; Batatia et al., 2022a;b) predominantly employs tensor products (TP) to preserve higher-order equivariant features. Tensor product operates on irreducible representations $\mathbf{u}$ of rotation order $\ell_1$ and $\mathbf{v}$ of rotation order $\ell_2$, yielding a new irreducible representation of order $\ell_3$ as

$$(\mathbf{u}^{\ell_1} \otimes \mathbf{v}^{\ell_2})_{m_3}^{\ell_3} = \sum_{m_1=-\ell_1}^{\ell_1} \sum_{m_2=-\ell_2}^{\ell_2} C_{(\ell_1,m_1),(\ell_2,m_2)}^{(\ell_3,m_3)} \mathbf{u}_{m_1}^{\ell_1} \mathbf{v}_{m_2}^{\ell_2}, \tag{2}$$

where $C$ denotes the Clebsch-Gordan (CG) coefficients (Griffiths and Schroeter, 2018) and $m \in \mathbb{N}$ denotes the $m$-th element in the irreducible representation. Here, $\ell_3$ satisfies $|\ell_1 - \ell_2| \leq \ell_3 \leq \ell_1 + \ell_2$, and $\ell_1, \ell_2, \ell_3 \in \mathbb{N}$. High-order equivariant 3D GNNs commonly use tensor products on the irreducible representations of neighbor nodes and edges to construct messages as

$$\mathbf{m}_{ij}^{\ell_o} = \sum_{\ell_i,\ell_f} R^{(\ell_i,\ell_f)}(\bar{r}_{ij})Y^{\ell_f}(\hat{r}_{ij}) \otimes \mathbf{x}_j^{\ell_i}, \tag{3}$$

where $|\ell_i - \ell_f| \leq \ell_o \leq \ell_i + \ell_f$, $\mathbf{x}_j$ denotes features of node $j$, and $R$ is a learnable non-linear function that takes the embedding of pairwise distance $\bar{r}_{ij}$ as input.

## 2.3 ATOMIC CLUSTER EXPANSION

Molecular potential and force field are crucial physical properties in molecular analysis. To approximate these properties, the atomic cluster expansion (ACE) (Drautz, 2019; Kovács et al., 2021) is used to approximate the atomic potential denoted as

$$E_i(\boldsymbol{\theta}_i) = \sum_j \sum_v c_v^{(1)} \phi_v(\mathbf{r}_{ij}) + \frac{1}{2} \sum_{j_1 j_2} \sum_{v_1 v_2} c_{v_1 v_2}^{(2)} \phi_{v_1}(\mathbf{r}_{ij_1}) \phi_{v_2}(\mathbf{r}_{ij_2})$$
$$+ \frac{1}{3!} \sum_{j_1 j_2 j_3} \sum_{v_1 v_2 v_3} c_{v_1 v_2 v_3}^{(3)} \phi_{v_1}(\mathbf{r}_{ij_1}) \phi_{v_2}(\mathbf{r}_{ij_2}) \phi_{v_3}(\mathbf{r}_{ij_3}) + \cdots, \tag{4}$$

where $\boldsymbol{\theta}_i = (\mathbf{r}_{ij_1}, \cdots, \mathbf{r}_{ij_N})$ denotes the $N$ bonds in the atomic environment, $\phi$ is the single bond basis function and $c$ is the coefficients. The computational complexity of modeling many-body potential increases exponentially with number of neighbors. To reduce the complexity, ACE further makes use of density trick to calculate the atomic energy via atomic base $A_{iv} = \sum_j \phi_v(\mathbf{r}_{ij})$, which has a linear complexity with the number of neighbors, denoted as

$$E_i(\boldsymbol{\theta}_i) = \sum_v c_v^{(1)} A_{iv} + \sum_{v_1 v_2}^{v_1 \geq v_2} c_{v_1 v_2}^{(2)} A_{iv_1} A_{iv_2} + \sum_{v_1 v_2 v_3}^{v_1 \geq v_2 \geq v_3} c_{v_1 v_2 v_3}^{(3)} A_{iv_1} A_{iv_2} A_{iv_3} + \cdots. \tag{5}$$

In this case, the computational complexity of modeling many-body interactions decreases to linear growth with the number of neighbors. With the reduced linearly complexity, many equivariant networks are designed to learn the many-body interactions. In these networks, spherical harmonic functions $Y(\hat{\mathbf{r}}_{ij})$ combined with radial functions $R(\bar{r}_{ij})$ are usually taken as the single bond basis function $\phi_v(\mathbf{r}_{ij})$. Then the aggregated atomic bases $A_{iv}$, typically represented as equivariant irreducible representations in these networks, are combined through tensor product operations to encode the many-body interactions while maintaining equivariance. Note that $v = 0, 1, 2, \ldots$ distinguishes among different basis. Specifically, BOTNet (Batatia et al., 2022a) takes multiple message passing layers to encode the many-body interaction and analyzes the body order for various message passing schemes. MACE (Batatia et al., 2022b) takes generalized Clebsch-Golden coefficients to couple the aggregated message to incorporate higher-order interactions. Allegro (Musaelian et al., 2023) uses a series of tensor product layers to calculate the equivariant representations without using message passing, learning many-body interactions.

## 2.4 UNIVERSALITY ANALYSIS

Universality is a powerful property for neural networks that can approximate arbitrary functions. While Zaheer et al. (2017); Maron et al. (2019); Keriven and Peyré (2019) study the universality of permutation invariant networks, several works have recently studied the rotational equivariant networks. Dym and Maron (2020) takes use of the proposed tensor representation to build $D$-spanning family and shows that Tensor Field Networks (TFN) (Thomas et al., 2018) is proved to be a universal equivariant network capable of approximating arbitrary equivariant functions defined on the point coordinates of point cloud data. Furthermore, GemNet (Gasteiger et al., 2021) uses the conclusion in Dym and Maron (2020), and is proved to be a universal GNN with directed edge embeddings and two-hop message passing.

## 3 THE PROPOSED PACE AND THEORETICAL ANALYSIS

### 3.1 EQUIVARIANT POLYNOMIAL FUNCTION APPROXIMATION

In this section, we first introduce the definitions of equivariant polynomial functions and their relationship to equivariant functions. Then we provide an analysis of the existing equivariant layer within the local atomic potential. Finally, we demonstrate our motivation to propose an equivariant message passing scheme to approximate higher-degree equivariant polynomial functions.

**Relationship between equivariant functions and equivariant polynomial functions.** Given a set of input 3D coordinates $\mathbf{C} = (c_1, c_2, \cdots, c_N)$ for $N$ nodes, a continuous equivariant function $\mathcal{C}_G(\mathbb{R}^{3 \times N}, W_T^N)$ maps these coordinates to equivariant features $W_T^N$, such as the irreducible node

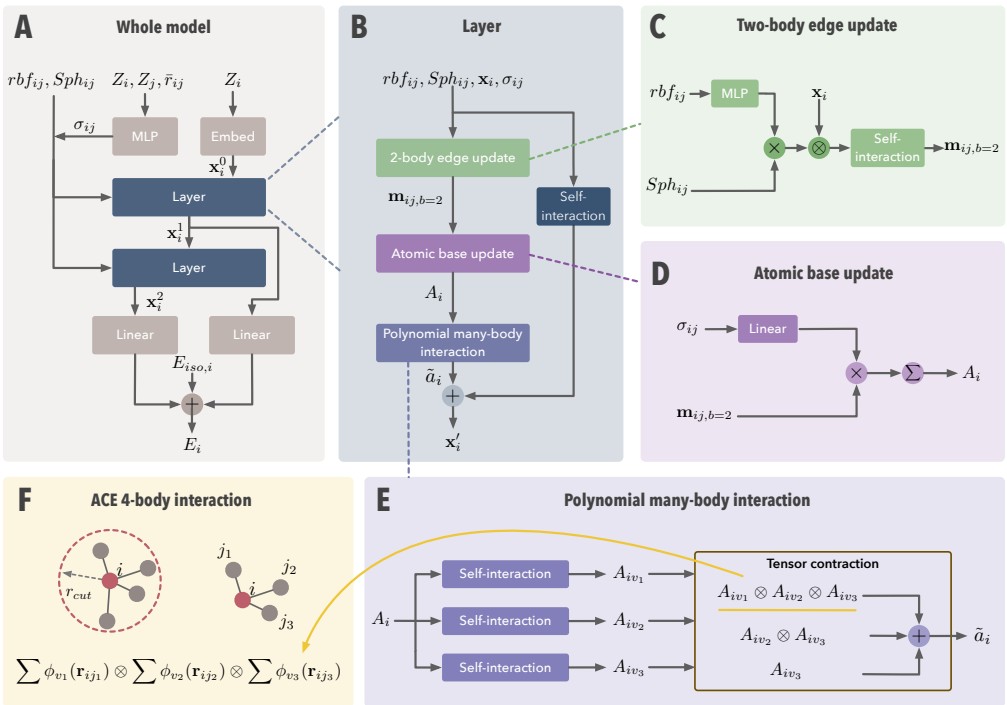

Figure 1: An architecture overview of PACE. **A:** The whole model. Inputs of PACE include atom types $Z$ and positions $\mathbf{C}$ based on which edge direction $\hat{\mathbf{r}}_{ij}$ and distance $\bar{r}_{ij}$ are calculated. The initial node features $\mathbf{x}_i^0$ is embedded using atom type. The spherical harmonics of edge direction $Sph_{ij}$, radius basis function transformed edge distance $rbf_{ij}$, and the invariant scaling features $\boldsymbol{\sigma}_{ij}$ are fed to each message passing layer. The readout block linearly transforms outputs of both message passing layers to predict local energies and then sum with isolated energies to obtain the total energy prediction. **B:** Message passing layer. Each message passing layer comprises a two-body edge update block, an atomic base update block, a polynomial many-body interaction block, and a skip connection with self-interaction. **C:** Two-body edge update block. $Sph_{ij}$ and MLP-transformed $rbf_{ij}$ are multiplied to produce the filter. Then, a tensor product is applied to the filter and node features, followed by a self-interaction to generate 2-body message. **D:** Atomic base update block. A linear transformation is applied to the invariant scaling edge features $\boldsymbol{\sigma}_{ij}$. Then, 2-body messages scaled by $\boldsymbol{\sigma}_{ij}$ are summed over neighboring nodes to form the atomic base for the central node. Same operation is proposed in Darby et al. (2023) named tensor sketch. **E:** Polynomial many-body interaction block. The atomic base $A_i$ is fed to multiple self-interaction layers separately to produce different $A_{iv}$. Then, tensor contraction is performed to produce $\tilde{a}_i$. **F:** An example of 4-body interaction in ACE. We aim to fit $\sum \phi_{v_1}(\mathbf{r}_{ij_1}) \otimes \sum \phi_{v_2}(\mathbf{r}_{ij_2}) \otimes \sum \phi_{v_3}(\mathbf{r}_{ij_3})$ using $A_{iv_1} \otimes A_{iv_2} \otimes A_{iv_3}$, where $\{\phi\}$ denotes the atomic base in ACE.

representations. This function maintains the desired property of equivariance with respect to the rotation, translation, and permutation operations defined by the group $G$. Furthermore, a $G$-equivariant polynomial $\mathcal{P}_G(\mathbb{R}^{3 \times N}, W_T^N)$ maps to equivariant features $W_T^N$ which are polynomial functions of the input coordinates. Lemma 1 from Dym and Maron (2020) demonstrates that any $\mathcal{C}_G(\mathbb{R}^{3 \times N}, W_T^N)$ can be uniformly approximated on compact sets by equivariant polynomials in $\mathcal{P}_G(\mathbb{R}^{3 \times N}, W_T^N)$. This implies that if equivariant networks can approximate all $G$-equivariant polynomial functions, they can further approximate any $G$-equivariant continuous functions. Inspired by this theorem, various geometric graph networks (Segol and Lipman, 2019; Gasteiger et al., 2021; Dym and Maron, 2020) provide analysis to demonstrate the capacity and expressive power of their networks in approximating equivariant polynomials, aiming to cover a broad range of equivariant polynomial functions.

**Motivation of Network Design.** Theorem 2 in Dym and Maron (2020) states that, if the input equivariant features can approximate $\mathcal{P}_G^D(\mathbf{C})$, which denotes any $G$-equivariant polynomial function with 3D coordinates $\mathbf{C}$ and the highest degree $D$, then by employing two tensor field network (TFN) layers with fully connected graphs, the output features can approximate $\mathcal{P}_G^{D+1}(\mathbf{C})$. While

TFN has the ability to approximate all equivariant polynomial functions with an infinite number of layers, the approximation capacity of a single TFN layer is limited, raising the need to develop modules improving the approximation capacity. To overcome this limitation, we propose a new equivariant graph network, known as PACE, to enhance the capacity of equivariant message passing to approximate higher-degree equivariant polynomials. In this work, instead of analyzing with polynomial functions based on atomic coordinates $\mathbf{C}$, the approximated polynomial functions focus on characterizing the atomic energy within the atomic environment, considering the N-bonds $\boldsymbol{\theta}_i$ in Equation 4. Through the theoretical analysis in section 3.3, a single message passing scheme in PACE can approximate any polynomial function $\mathcal{P}_G^{D=v}(\boldsymbol{\theta}_i)$, where $v$ is the number of bases in the polynomial many-body interaction module.

**Analysis of existing equivariant networks.** For existing equivariant networks in molecular property prediction (Batzner et al., 2022; Musaelian et al., 2023; Batatia et al., 2022b), we also provide theoretical justification about their ability to approximate polynomial functions focusing on atomic environment, considering the N-bonds $\boldsymbol{\theta}_i$. As shown in Table 1, a single NequIP layer can approximate polynomial function $\mathcal{P}^{D=1}(\boldsymbol{\theta}_i)$ as shown in A.2.5, Allegro can approximate polynomial function $\mathcal{P}_G^{D=N_{\text{layer}}}(\boldsymbol{\theta}_i)$ as discussed in A.2.7, and our proposed PACE can approximate $\mathcal{P}^{D=v}(\boldsymbol{\theta}_i)$ with a single layer with Theorem 2. For the MACE model, we discuss it in the appendix A.2.5.

Table 1: A comparison of the ability to approximate polynomial function for various equivariant architecture within local atomic bonds in the atomic environment.

| Network Architecture | Highest Degree in Polynomial $\mathcal{P}^D(\boldsymbol{\theta}_i)$ |
|---|---|
| NequIP layer | $D = 1$ |
| Allegro | $D = N_{\text{layer}}$ |
| PACE layer | $D = v$ |

## 3.2 MODEL ARCHITECTURE

In this subsection, we introduce the architectural details of the proposed PACE model, including the embedding layer, message passing layer, and output layer.

### 3.2.1 INPUT EMBEDDING

The input to PACE consists of atom types $Z \in \mathbb{N}^{N \times 1}$ and corresponding positions $\mathbf{C} \in \mathbb{R}^{N \times 3}$, where $N$ represents the number of atoms in a molecule. Edges are constructed based on a cutoff distance $r_{cut}$. The node features $\mathbf{x}_i^0$ are initialized through a linear transformation applied to its atomic type. Edge orientations are denoted by spherical harmonics $Y^\ell(\hat{\mathbf{r}}_{ij})$, and pairwise distances $\bar{r}_{ij}$ are embedded using Bessel functions with a smoothed polynomial cutoff (Gasteiger et al., 2020). Besides, we introduce invariant scaling features $\boldsymbol{\sigma}_{ij}$ for each edge by

$$\boldsymbol{\sigma}_{ij} = \text{MLP}(\text{one-hot}(Z_i) \| \text{one-hot}(Z_j)), \tag{6}$$

where MLP is a multiple layer perceptron and one-hot$(\cdot)$ is one-hot encoding of atom type. The scaling edge features $\boldsymbol{\sigma}_{ij}$ are used for message aggregation in each layer.

### 3.2.2 MESSAGE PASSING LAYER

Each layer of the proposed PACE is comprised of four blocks that sequentially perform 2-body edge update, 3-body update, atomic base update, and polynomial many-body interaction. Finally, the updated node features are skip-connected with the self-interaction transformed input node features and used as output. We describe the layer architecture below and provide an illustration in Figure 1.

**Two-Body Edge Update.** The messages from neighboring nodes to the central node are typically determined by node features, edge orientation, and distance between two nodes. As illustrated in Figure 1 C, we first construct a filter based on edge orientation and distance as

$$F^{(\ell_i, \ell_f)}(\bar{r}_{ij}, \hat{\mathbf{r}}_{ij}) = R^{(\ell_i, \ell_f)}(\bar{r}_{ij}) Y^{\ell_f}(\hat{\mathbf{r}}_{ij}). \tag{7}$$

Then, a tensor product is applied to the filter and irreducible representations of node $j$ to produce the message $\mathbf{m}_{ij,b=2}$ from node $j$ to node $i$ as

$$\mathbf{m}_{ij,b=2}^{\ell_o} = \sum_{\ell_i, \ell_f} F^{(\ell_i, \ell_f)}(\bar{r}_{ij}, \hat{\mathbf{r}}_{ij}) \otimes \mathbf{x}_j^{\ell_i}. \tag{8}$$

Finally, $\mathbf{m}_{ij,b=2}$ is used as output after a self-interaction. Similar to messages in a standard message passing framework, the message obtained here has a body order of 2 because it involves the interaction of only two atoms.

**Atomic Base Update.** In the original ACE method, the atomic base of the central node is constructed by summing over one-particle base functions that are analogous to messages from neighboring nodes. In our method, we scale equivariant messages using linearly transformed invariant scaling features $\boldsymbol{\sigma}_{ij}$, and use these to generate the atomic base $A_i$ as

$$A_i = \frac{1}{|\mathcal{N}(i)|} \sum_{j \in \mathcal{N}(i)} W \boldsymbol{\sigma}_{ij} \mathbf{m}_{ij,b=3}, \quad (9)$$

where $|\mathcal{N}(i)|$ is the number of neighbor nodes and $W$ is the weight matrix. Operations in this block are shown in Figure 1 D.

**Polynomial Many-Body Interaction.** To incorporate many-body interactions, the polynomial many-body interaction module is proposed to mix the atomic base $A_i$. As shown in Figure 1 E, self-interaction layers are used to map the input atomic base $A_i$ to different bases, distinguishing the atomic bases for different body orders. Then tensor contraction (Batatia et al., 2022b) uses generalized Clebsch-Golden to fuse multiple atomic bases. For example, when fusing two irreps, Clebsch-Gordan coefficients $C^{\ell_3 m_3}_{\ell_1 m_1, \ell_2 m_2}$ are used to maintain equivariance when fusing two irreps with rotation orders $\ell_1$ and $\ell_2$ to the output $\ell_3$, as shown in Equation 2, and the triplet $(\ell_1, \ell_2, \ell_3)$ is defined as a path. In general, when fusing $N$ irreps, the generalized Clebsch-Gordan coefficients are used to maintain the equivariance, defined as

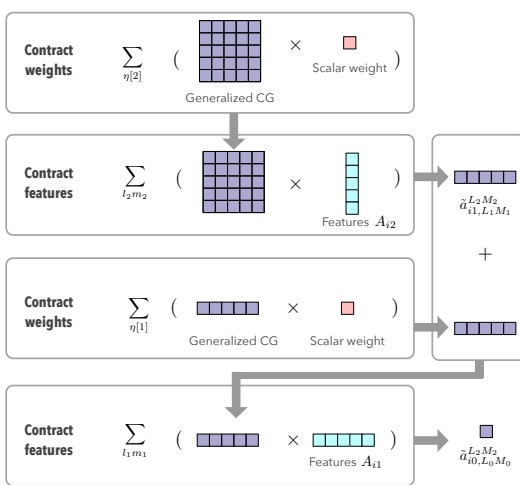

Figure 2: Illustration of tensor contraction in the polynomial many-body interaction module. This figure demonstrates an example of 3-body interactions with $v = 2$ and final $L_2 = 0, M_2 = 0$. Note that the contract weights operation learns weighted summation over all paths $\eta[v]$, where $\eta[v] = (\ell_1, \ell_2, L_2, \cdots, \ell_v, L_v)$.

$$\mathcal{C}^{\mathcal{L}[N]\mathcal{M}[N]}_{\ell_1 m_1, \ldots, \ell_n m_n} = C^{L_2 M_2}_{\ell_1 m_1, \ell_2 m_2} C^{L_3 M_3}_{L_2 M_2, \ell_3 m_3} \cdots C^{L_N M_N}_{L_{N-1} M_{N-1}, \ell_N m_N}, \quad (10)$$

where $\mathcal{L}[N] = (\ell_1, L_2, \cdots L_N)$ with $|L_{i-1} - \ell_i| \le L_i \le |L_{i-1} + \ell_i|, L_i \in \mathbb{N}, \forall i \ge 2, i \in \mathbb{N}_+$, and the path is shown as $\eta[N] = (\ell_1, \ell_2, L_2, \ell_3, L_3, \cdots, \ell_{N-1}, L_{N-1}, \ell_N, L_N)$. Then the output irreps is contracted one by one to consider the coupled many-body interactions shown as

$$\tilde{a}^{L_N M_N}_{i(v-1), L_{v-1} M_{v-1}} = \sum_{\ell_v} \sum_{m_v=-\ell_v}^{\ell_v} A_{iv,l_v m_v} \sum_{\eta[v]} \left( W_{v,\eta} * \mathcal{C}^{\mathcal{L}[v]\mathcal{M}[v]}_{\ell_1 m_1, \ldots, \ell_v m_v} + \tilde{a}^{L_N M_N}_{iv, L_v M_v} \right), \quad (11)$$

where $W$ is the path weight, $\tilde{a}$ is the intermediate irreps, and $v \in \mathbb{N}_+$ starts from $N$ to 1 to incorporate N-body interactions. Note that $\mathcal{C} = 1$ and $L_0 = 0$ when $v = 1$, and $\tilde{a} = 0$ when $v = N$. Compared to the higher order features in MACE, the implementation of contraction is the same, but we take multiple self-interaction layers to distinguish the atomic base from the same $A_i$ to different $A_{iv}$. Thus, our proposed message passing can achieve complete equivariant polynomial approximation. We illustrate tensor contraction in Figure 2 and Figure 4.

### 3.2.3 OUTPUT

We follow (Batatia et al., 2022b) to extract and transform the invariant part of node features produced by each layer to compute the local energy of node $i$ as

$$E_i = E_{iso,i} + W \mathbf{x}^1_{i,00} + \text{MLP}(\mathbf{x}^2_{i,00}), \quad (12)$$

where $E_{iso,i}$ denotes the isolated energy corresponding to the atom type of node $i$, which is a known value. $W$ denotes a linear function and MLP is a multiple layer perceptron. The total energy of the molecule is the sum of local energies. Once the total energy is predicted, we then use $\mathbf{f}_i = -\frac{\partial E}{\partial \mathbf{C}_i}$ to calculate the force acting on each atom, as it ensures energy conservation.

### 3.3 THEORETICAL STUDIES

In this section, we provide a theoretical analysis elucidating how our model enhances the highest degree of the approximated polynomial functions. To facilitate the analysis, we employ a powerful tool known as tensor representation, which leverages a series of matrix kronecker product on the input directions. Specifically, the tensor representation of two directions $r_{ij_1}, r_{ij_2} \in \mathbb{R}^3$ can be denoted as $r_{ij_1} \otimes r_{ij_2} \in \mathbb{R}^{3\times3}$. Moreover, the tensor representation resulting from applying the kronecker product to $r_{ij}$ for $k$ times is denoted as $r_{ij}^{\otimes k} \in \mathbb{R}^{3^k}$. Note that tensor representation exhibits SO(3) equivariance owning to the SO(3) equivariant nature of tensor product operation (Thomas et al., 2018). Examples of tensor representation are illustrated in Figure 3.

Following Theorem 1 in Dym and Maron (2020), there are two necessary conditions when an equivariant network can approximate any polynomial equivariant function with the highest degree $D$. Firstly, the network's features $\mathcal{F}_{\text{feat}}$ must be D-spanning. Secondly, the linear pooling layer $\mathcal{F}_{\text{pool}}$ in this network must be linear universal. As demonstrated in Lemma 5 of (Dym and Maron, 2020), the self-interaction layer is linear universal since it covers all linear mappings between irreducible representations. Hence, it can be concluded that an equivariant graph network can approximate all the equivariant polynomials with the highest degree $D$, as long as its irreducible representation is D-spanning. Here, we precisely recall the definition of D-spanning as follows:

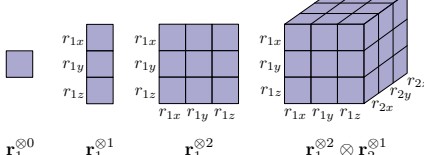

Figure 3: Examples of tensor representation. The tensor representation, denoted as $\mathbf{r}_1^{\otimes t_1} \otimes \cdots \otimes \mathbf{r}_n^{\otimes t_n}$, has a shape of $\mathbb{R}^{3^{T_n}}$, where $T_n = \sum_{i=1}^{n} t_i$.

**Definition 1.** *(D-spanning). For $D \in \mathbb{N}_+$, let $\mathcal{F}_{feat}$ be a subset of $\mathcal{C}_G(\mathbb{R}^{3\times N}, W_{feat}^N)$. We say that $\mathcal{F}_{feat}$ is D-spanning, if there exist $f_1, \cdots, f_K \in \mathcal{F}_{feat}$, such that every polynomial $\mathbb{R}^{3\times N} \to \mathbb{R}^N$ of degree $D$ which is invariant to translations and equivariant to permutations, can be written as $p(X) = \sum_{k=1}^{K} \hat{\Lambda}_k (f_k(X))$, where $\Lambda_k : W_{feat} \to \mathbb{R}$ are all linear functionals, and $\hat{\Lambda}_k : W_{feat} \to \mathbb{R}$ are the functions defined by element-wise applications of $\Lambda_k$.*

Based on atomic cluster expansion (ACE), the atomic energy function is defined as

$$E_i(\boldsymbol{\theta}_i) = E_i(\mathbf{r}_{i1}, \mathbf{r}_{i2}, \ldots, \mathbf{r}_{iN}) \tag{13}$$

We extend the D-spanning function introduced in Dym and Maron (2020) considering the N-bonds $\boldsymbol{\theta}_i$ in atomic environment, defined as

$$Q_K^{(\mathbf{t})}(\boldsymbol{\theta}_i) = \sum_{j_1, j_2, \ldots, j_K = 1}^{N} \mathbf{r}_{ij_1}^{\otimes t_1} \otimes \mathbf{r}_{ij_2}^{\otimes t_2} \otimes \mathbf{r}_{ij_3}^{\otimes t_3} \otimes \ldots \otimes \mathbf{r}_{ij_K}^{\otimes t_K}, \tag{14}$$

where $\mathbf{t} = (t_1, \cdots, t_K)$. Then, the polynomial function set $Q_K^D$

$$Q_K^D = \left\{ \iota \circ Q_K^{(\mathbf{t})}(\boldsymbol{\theta}_i) \mid \|\boldsymbol{t}\|_1 \le D \right\}, \tag{15}$$

which is a D-spanning family when $K \ge D$, and $\iota$ denotes a function mapping from the tensor representation to equivariant features $W_T^N$. We further build a connection between $Q^{(\mathbf{t})}(\boldsymbol{\theta}_i)$ and the irreducible representations (irreps) used by networks shown in Theorem 1. A detailed proof of this relationship is provided in Appendix A.1.

**Theorem 1.** *For any D-spanning function $Q_K^{(\mathbf{t})}(\boldsymbol{\theta}_i)$ appeared in $Q_K^D$ and for any position $P = (p_1, p_2, \cdots, p_k)$ in tensor representation, where $p_k \in \mathbb{R}^3$ denotes the element position, if there exists $w_1$ and $irreps_1$ such that $Q_K^{(\mathbf{t})}(\boldsymbol{\theta}_i)(P) = \sum_{\ell m} w_1^{\ell m} irreps_1^{\ell m}$, then we say that the irreducible representation $irreps_1$ represents $Q_K^{(\mathbf{t})}(\boldsymbol{\theta}_i)$, and the set of $irreps_1$ forms a D-spanning family.*

In Appendix A.2.2, we provide a detailed proof to show that the irreps outputted by our two-body edge update block can represent $Q^{(t_1)}(\boldsymbol{\theta}_i) = \sum_{j_1=1}^{N} \mathbf{r}_{ij_1}^{\otimes t_1}$, where $t_1 \le D$. Moreover, based on Theorem 2 and the detailed proof in Appendix A.2.3, we show that the irreps outputted by our polynomial many-body interaction block can represent $Q^{(\mathbf{t}_1, \mathbf{t}_2, \cdots, \mathbf{t}_v)}(\boldsymbol{\theta}_i) = \sum_{j_1, j_2, \cdots, j_v = 1}^{N} \mathbf{r}_{ij_1}^{\otimes t_1} \otimes \mathbf{r}_{ij_2}^{\otimes t_2} \otimes \cdots \otimes \mathbf{r}_{ij_v}^{\otimes t_v}$. Therefore, as the output irreps of PACE belong to the D-spanning family with $D = v$, a single PACE layer can effectively approximate any polynomial function with the highest degree of $D = v$.

**Theorem 2.** *If the input irreps can represent $Q^{(\mathbf{t})}(\boldsymbol{\theta}_i)$ with $\|\mathbf{t}\|_1 \leq v$, the proposed polynomial many-body interaction module can approximate $Q^{(\mathbf{t}_1, \mathbf{t}_2, \cdots, \mathbf{t}_v)}(\boldsymbol{\theta}_i)$ for any $\|\mathbf{t}\|_1 \leq v$, where $v$ is the number of bases in the polynomial MB interaction modules.*

## 4 EXPERIMENTS

We conduct experiments on two molecular dynamics simulation datasets, the revised MD17 (rMD17) and 3BPA datasets. The proposed PACE is trained using these datasets to predict both the invariant energy of the entire molecule and the equivariant forces acting on individual atoms. Among our baselines (Kovács et al., 2021; Christensen et al., 2020; Bartók et al., 2010; Smith et al., 2017; Gasteiger et al., 2021; Batzner et al., 2022; Batatia et al., 2022a; Musaelian et al., 2023; Batatia et al., 2022b), NequIP, BOTNet, Allegro and MACE are all equivariant graph neural networks (GNNs) with rotation order $\ell > 1$. In particular, BOTNet, Allegro, and MACE incorporate many-body interactions, while ACE is a parameterized physical model that does not belong to the class of neural networks. Our experiments are implemented with PyTorch 1.11.0 (Paszke et al., 2019), PyTorch Geometric 2.1.0 (Fey and Lenssen, 2019), and e3nn (Geiger and Smidt, 2022). In experiments, we train models on a single 11GB Nvidia GeForce RTX 2080Ti GPU and Intel Xeon Gold 6248 CPU.

### 4.1 THE rMD17 DATASET

**Dataset.** The rMD17 (Christensen and Von Lilienfeld, 2020) is a benchmark dataset that comprises ten small organic molecular systems. Each molecule in the dataset is accompanied by 1000 3D structures, which were generated through meticulously accurate ab initio molecular dynamic simulations employing density functional theory (DFT). These structures capture the diverse conformational space of the molecules and are valuable for studying their quantum properties.

**Setup.** In our experiments, we use officially provided random splits. Next, we use the same splitting seed as MACE to further divide the training set into a training set comprising 950 structures and a validation set comprising 50 structures. Then, we perform our evaluations on the test set with 1000 structures. Training details are provided in Appendix C.

**Results.** Table 2 summarizes the performance of our proposed method in comparison to baselines on all ten molecules in the rMD17 dataset. Mean absolute errors (MAE) are employed as the evaluation metric for both energy and force predictions. It is worth noting that our PACE demonstrates state-of-art performance in energy prediction across all molecules. Specifically, we achieved significant improvements of 33.3%, 33.3% and 25.0% on Benzene, Toluene, and Ethanol, respectively. In terms of force prediction, PACE achieves state-of-the-art performance on eight out of the ten molecules, exhibiting substantial improvements of 20.0% on Toluene and 11.5% on Azobenzene, respectively. Besides, we attain the second-best on the other two molecules.

### 4.2 THE 3BPA DATASET

**Dataset.** The 3BPA dataset (Kovács et al., 2021) is also generated through molecular dynamic simulations employing Density Functional Theory (DFT). Unlike rMD17, this dataset is specifically focused on a single flexible molecule, namely the 3BPA molecule. 3BPA is characterized by three freely rotating angles, which primarily induce structural changes at varying temperatures. As a result, 3BPA is frequently employed to assess the generalization capability of methods when confronted with out-of-distribution test sets.

**Setup.** Our model is trained using a training set consisting of 450 structures and a validation set comprising 50 structures. Both the training and validation sets were sampled at 300K. Then, the performance of the model was assessed on three distinct test sets that are sampled at three different temperatures: 300K, 600K, and 1200K. We provide training details in Appendix C.

**Results.** Table 3 summarizes the performance of our proposed method in the 3BPA dataset. Here, root-mean-square error (RMSE) is used as the evaluation metric. The proposed PACE shows comparable performance to MACE, while outperforming other baseline methods significantly.

Table 2: Performance on the rMD17 dataset. Mean absolute errors (MAE) are reported for both energy (E) and force (F) predictions, with meV and meV/Å as units, respectively. Bold numbers highlight the best performance.

| | | ACE | FCHL | GAP | ANI | GemNet (T/Q) | NequIP | BOTNet | Allegro | MACE | Ours |
|---|---|---|---|---|---|---|---|---|---|---|---|
| Aspirin | E | 6.1 | 6.2 | 17.7 | 16.6 | - | 2.3 | 2.3 | 2.3 | 2.2 | **1.8** |
| | F | 17.9 | 20.9 | 44.9 | 40.6 | 9.5 | 8.2 | 8.3 | 7.3 | 6.6 | **6.0** |
| Azobenzene | E | 3.6 | 2.8 | 8.5 | 15.9 | - | 0.7 | 0.7 | 1.2 | 1.2 | **0.6** |
| | F | 10.9 | 10.8 | 24.5 | 35.4 | - | 2.9 | 3.3 | 2.6 | 3.0 | **2.3** |
| Benzene | E | 0.04 | 0.35 | 0.75 | 3.3 | - | 0.04 | 0.03 | 0.4 | 0.4 | **0.02** |
| | F | 0.5 | 2.6 | 6.0 | 10.0 | 0.5 | 0.3 | 0.3 | **0.2** | 0.3 | 0.2 |
| Ethanol | E | 1.2 | 0.9 | 3.5 | 2.5 | - | 0.4 | 0.4 | 0.4 | 0.4 | **0.3** |
| | F | 7.3 | 6.2 | 18.1 | 13.4 | 3.6 | 2.8 | 3.2 | 2.1 | 2.1 | **2.0** |
| Malonaldehyde | E | 1.7 | 1.5 | 4.8 | 4.6 | - | 0.8 | 0.8 | **0.6** | 0.8 | **0.6** |
| | F | 11.1 | 10.3 | 26.4 | 24.5 | 6.6 | 5.1 | 5.8 | **3.6** | 4.1 | 3.9 |
| Naphthalene | E | 0.9 | 1.2 | 3.8 | 11.3 | - | 0.9 | **0.2** | **0.2** | 0.5 | **0.2** |
| | F | 5.1 | 6.5 | 16.5 | 29.2 | 1.9 | 1.3 | 1.8 | **0.9** | 1.6 | **0.9** |
| Paracetamol | E | 4.0 | 2.9 | 8.5 | 11.5 | - | 1.4 | 1.3 | 1.5 | 1.3 | **1.0** |
| | F | 12.7 | 12.3 | 28.9 | 30.4 | - | 5.9 | 5.8 | 4.9 | 4.8 | **4.4** |
| Salicylic acid | E | 1.8 | 1.8 | 5.6 | 9.2 | - | 0.7 | 0.8 | 0.9 | 0.9 | **0.5** |
| | F | 9.3 | 9.5 | 24.7 | 29.7 | 5.3 | 4.0 | 3.1 | 4.3 | **2.9** | **2.9** |
| Toluene | E | 1.1 | 1.7 | 4.0 | 7.7 | - | 0.3 | 0.3 | 0.4 | 0.5 | **0.2** |
| | F | 6.5 | 8.8 | 17.8 | 24.3 | 2.2 | 1.6 | 1.9 | 1.8 | 1.5 | **1.2** |
| Uracil | E | 1.1 | 0.6 | 3.0 | 5.1 | - | 0.4 | 0.4 | 0.6 | 0.5 | **0.3** |
| | F | 6.6 | 4.2 | 17.6 | 21.4 | 3.8 | 3.1 | 3.2 | **1.8** | 2.1 | 2.0 |

Table 3: Performance on the 3BPA dataset. Root-mean-square errors (RMSE) are reported for both energy (E) and force (F) predictions, with meV and meV/Å as units, respectively. Standard deviations are calculated over three runs with different seeds. Bold numbers highlight the best performance.

| | | ACE | NequIP | BOTNet | Allegro | MACE | Ours |
|---|---|---|---|---|---|---|---|
| 300K | E | 1.1 | 3.28 (0.12) | 3.1 (0.13) | 3.84 (0.10) | 3.0 (0.2) | **2.5** (0.2) |
| | F | 27.1 | 10.77 (0.28) | 11.0 (0.14) | 12.98 (0.20) | **8.8** (0.3) | 9.1 (0.1) |
| 600K | E | 24.0 | 11.16 (0.17) | 11.5 (0.6) | 12.07 (0.55) | 9.7 (0.5) | **9.6** (0.1) |
| | F | 64.3 | 26.37 (0.11) | 26.7 (0.29) | 29.11 (0.27) | **21.8** (0.6) | 22.1 (0.1) |
| 1200K | E | 85.3 | 38.52 (2.00) | 39.1 (1.1) | 42.57 (1.79) | 29.8 (1.0) | **29.0** (0.7) |
| | F | 187.0 | 76.18 (1.36) | 81.1 (1.5) | 82.96 (2.17) | 62.0 (0.7) | **61.6** (0.3) |

## 4.3 MORE EMPIRICAL ANALYSES

In addition to rMD17 and 3BPA datasets, we also perform experiments on another molecular dataset, AcAc, and show results in Appendix D.1. To further evaluate the ability of PACE to simulate realistic molecular dynamics (MD), we conduct MD simulations and analyze MD trajectories generated by the trained PACE model. Details and results are presented in Appendix D.2. Moreover, we perform an ablation study to analyze the effectiveness of the polynomial many-body interaction module, which is the key component of PACE. Settings and results are described in Appendix D.3. Furthermore, we provide a comparison of algorithm efficiency between our PACE and baseline methods in Appendix D.4.

## 5 CONCLUSION

In this work, we introduced PACE, a new equivariant network for atomic potential and force field predictions. PACE has been meticulously designed to consider many-body interactions based on the principles of the Atomic Cluster Expansion (ACE) mechanism. The message passing scheme employed in PACE is unique in its sound physical foundations and its capability to approximate any equivariant polynomial functions with constrained degree. The comprehensive experimental results and detailed analyses, encompassing energy and force predictions as well as Molecular Dynamics simulations, provide compelling evidence for the efficacy of the proposed PACE model, which is further supported by our comprehensive theoretical analysis.

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

# A  THEORETICAL PROOF

## A.1  PROOF OF THEOREM 1

*Proof.* Since $Q_K^D$ is a D-spanning family, then there exists $f_1, \cdots, f_K \in Q_K^{(\mathbf{t})}$ with $\|\mathbf{t}\|_1 \leq D$, that each polynomial function $p$ with the highest degree no more than $D$ can be represented as the linear combination with linear pooling function $\hat{\Lambda}_k$ on them

$$p = \sum_k w_k * \hat{\Lambda}_k(f_k) = \sum_k w_k \hat{\Lambda}_k(Q_K^{(\mathbf{t_k})}(\boldsymbol{\theta}_i)) \tag{16}$$

$$= \sum_k w_k \sum_P W_P^* Q_K^{(\mathbf{t_k})}(\boldsymbol{\theta}_i)(P), \tag{17}$$

where $P = (p_1, p_2, \cdots, p_K)$ is the position of the entry, and $W_P^*$ is the corresponding weight.

Since $Q^{(\mathbf{t_k})}(\boldsymbol{\theta}_i)(P) = \sum_{\ell m} w_1^{\ell m} \text{irreps}_1^{\ell m}$, the $p$ can be represented as

$$p = \sum_k w_k \sum_P W_P^* \sum_{\ell m} w_{1,\mathbf{t_k},P}^{\ell m} \text{irreps}_{1,\mathbf{t_k},P}^{\ell m} = \sum_k w_k \sum_{\ell m P} w_{1,\mathbf{t_k},P}^{*\ell m} \text{irreps}_{1,\mathbf{t_k},P}^{\ell m} \tag{18}$$

where the $\text{irreps}_{1,\mathbf{t_k},P}^{\ell m}$ can be obtained by various channels and $w_{1,\mathbf{t_k},P}^{*\ell m} = w_{1,\mathbf{t_k},P}^{\ell m} W_P^*$. Therefore, the set of $\text{irreps}_1$ is D-spanning. □

## A.2  PROOF OF D-SPANNING IRREPS IN PACE

### A.2.1  LEMMAS

**Lemma 1.** *If $\text{irreps}_1$ can represent $Q^{\mathbf{t_1}}(\boldsymbol{\theta}_i)$, $\text{irreps}_2$ can represent $Q^{\mathbf{t_2}}(\boldsymbol{\theta}_i)$ and their tensor product output $\text{irreps}_3$ can represent $Q^{(\mathbf{t_1},\mathbf{t_2})}(\boldsymbol{\theta}_i)$.*

*Proof.* After two-body edge update module and the atomic base update modules, $Q^{\mathbf{t_1}} = \sum_{j_1} \mathbf{r}_{ij}^{\otimes t_1}$ for tensor representation format, and the value at position $P_1 = (p_{11}, \cdots, p_{iL})$ can be represented by linear combination of the elements in $\text{irreps}_1$, shown as

$$Q^{\mathbf{t_1}}(P_1) = \sum_{\ell_1 m_1} w_1^{\ell_1 m_1} \text{irreps}_1^{\ell_1 m_1}, \tag{19}$$

When the tensor product of $\text{irreps}_1$ and $\text{irreps}_2$ is $\text{irreps}_3$, the representation of $\text{irreps}_3$ is denoted as

$$\text{irreps}_3^{\ell_3(\ell_1,\ell_2)m_3} = \sum_{m_1 m_2} C_{\ell_1 m_1, \ell_2 m_2}^{\ell_3 m_3} \text{irreps}_1^{\ell_1 m_1} \text{irreps}_2^{\ell_2 m_2} \tag{20}$$

When the $Q^{\mathbf{t_2}}(P_2)$ can be linear combination of elements in $\text{irreps}_2$, then

$$Q^{(\mathbf{t_1},\mathbf{t_2})}(P_1, P_2) = \left(\sum_{\ell_1 m_1} w_1^{\ell_1 m_1} \text{irreps}_1^{\ell_1 m_1}\right)\left(\sum_{\ell_2 m_2} w_2^{\ell_2 m_2} \text{irreps}_2^{\ell_2 m_2}\right)$$

$$= \sum_{\ell_1 m_1 \ell_2 m_2} w_1^{\ell_1 m_1} w_2^{\ell_2 m_2} \text{irreps}_1 \text{irreps}_2 \tag{21}$$

Then when the linear combination of the $\text{irreps}_3$ is shown as

$$Q'^{(\mathbf{t_1},\mathbf{t_2})}(P_1, P_2) = \sum_{\ell_3(\ell_1,\ell_2)m_3} w_3^{\ell_3(\ell_1,\ell_2)m_3} \text{irreps}_3^{\ell_3(\ell_1,\ell_2)m_3}$$

$$= \sum_{\ell_3(\ell_1,\ell_2)m_3} w_3^{\ell_3(\ell_1,\ell_2)m_3} \sum_{\ell_1 m_1 \ell_2 m_2} C_{\ell_1 m_1, \ell_2 m_2}^{\ell_3 m_3} \text{irreps}_1^{\ell_1 m_1} \text{irreps}_2^{\ell_2 m_2}$$

$$= \sum_{\ell_1 m_1 \ell_2 m_2} \text{irreps}_1^{\ell_1 m_1} \text{irreps}_2^{\ell_2 m_2} \sum_{l_3(\ell_1,\ell_2)m_3} C_{\ell_1 m_1, \ell_2 m_2}^{\ell_3 m_3} w_3^{\ell_3(\ell_1,\ell_2)m_3} \tag{22}$$

When $w_3^{\ell_3(\ell_1,\ell_2)m_3} = \sum_{\ell_1 m_1,\ell_2 m_2} C_{\ell_1 m_1,\ell_2 m_2}^{\ell_3 m_3} w_1^{\ell_1 m_1} w_2^{\ell_2 m_2}$, we have

$$
\sum_{\ell_1 m_1,\ell_2 m_2} C_{\ell_1 m_1,\ell_2 m_2}^{\ell_3 m_3} w_3^{\ell_3(\ell_1,\ell_2)m_3}
$$

$$
= \sum_{\ell_1 m_1,\ell_2 m_2} C_{\ell_1 m_1,\ell_2 m_2}^{\ell_3 m_3} \sum_{\ell_3(\ell_1,\ell_2)m_3} C_{\ell_1 m_1,\ell_2 m_2}^{\ell_3 m_3} w_1^{\ell_1 m_1} w_2^{\ell_2 m_2}
$$

$$
= \sum_{\ell_1 m_1,\ell_2 m_2} w_1^{\ell_1 m_1} w_2^{\ell_2 m_2} \sum_{\ell_3(\ell_1,\ell_2)m_3} \left( C_{\ell_1 m_1,\ell_2 m_2}^{\ell_3 m_3} C_{\ell_1 m_1,\ell_2 m_2}^{\ell_3 m_3} \right)
$$

$$
= \sum_{\ell_1 m_1,\ell_2 m_2} w_1^{\ell_1 m_1} w_2^{\ell_2 m_2} \tag{23}
$$

Therefore,

$$
Q^{'(\mathbf{t_1},\mathbf{t_2})}(P_1,P_2) = \sum_{\ell_1 m_1,\ell_2 m_2} \mathrm{irreps}_1^{\ell_1 m_1} \mathrm{irreps}_2^{\ell_2 m_2} \sum_{\ell_1 m_1,\ell_2 m_2} w_1^{\ell_1 m_1} w_2^{\ell_2 m_2}
$$

$$
= \sum_{\ell_1 m_1,\ell_2 m_2} \mathrm{irreps}_1^{\ell_1 m_1} \mathrm{irreps}_2^{\ell_2 m_2} w_1^{\ell_1 m_1} w_2^{\ell_2 m_2}
$$

$$
= Q^{(\mathbf{t_1},\mathbf{t_2})}(P_1,P_2) \tag{24}
$$

Above all, the $\mathrm{irreps}_3$ can represent $Q^{(\mathbf{t_1},\mathbf{t_2})}(P_1,P_2)$. □

### A.2.2 PROOF OF D-SPANNING IRREPS FOR TWO-BODY EDGE UPDATE MODULE

*Proof.* With edge features in $\mathbf{r}_{ij}^{\otimes t_1}$ with $\|\mathbf{t}_1\|_1 \leq v$, then the features after summation over the neighbors is denoted as $Q^{\mathbf{t_1}}(\boldsymbol{\theta}_i) = \sum_j \mathbf{r}_{ij}^{\otimes \mathbf{t_1}}$. With spherical harmonics with suitable $L_{\max}$, it can achieve $\|\mathbf{t}_1\| = v$. □

### A.2.3 PROOF OF THEOREM 2

Consider the two body edge update module, the path can be represented as $\eta[2] = (\ell_1, \ell_2, L_2)$, we have $\mathrm{irreps}_1$ and $\mathrm{irreps}_2$ to represent $Q^{\mathbf{t_1}}(P_1)$ and $Q^{\mathbf{t_2}}(P_2)$ with $\|\mathbf{t}_1\|_1 \leq v, \|\mathbf{t}_2\|_1 \leq v$, respectively. Then with Lemma 1, the output irreps can represent $Q^{(\mathbf{t_1},\mathbf{t_2})}(P_1,P_2)$. Note that there might be multiple channels for the same rotation order $L_2$, and we use $L_2(\ell_1,\ell_2)$ to distinguish them. Meanwhile, the weighted sum over these equivariant irreducible representations can also achieve representativity.

For the path $\eta[v] = (\ell_1, \ell_2, L_2, \cdots, \ell_v, L_v)$, the $\mathrm{irreps}_{\eta[v-1]}$ can represent $Q^{(\mathbf{t_1},\mathbf{t_2},\cdots,\mathbf{t}_{v-1})}$, and the $\mathrm{irreps}_v$ can represent $Q^{\mathbf{t}_v}$.

$$
\mathrm{irreps}_3^{L_v' M_v} = \sum_{M_{v-1} m_v} C_{L_{v-1}' M_{v-1}, \ell_v m_v}^{L_v M_v} \mathrm{irreps}_1^{L_{v-1}' M_{v-1}} \mathrm{irreps}_2^{\ell_v m_v}, \tag{25}
$$

where $L_v' = L_v(\eta[v-1], \ell_v)$ and $L_{v-1}' = L_{v-1}(\eta[v-2], \ell_{v-1})$. Then, in this case, we extend the proof of Lemma 1 and prove the

$$
Q^{'(\mathbf{t_1},\mathbf{t_2},\mathbf{t}_v)}(P_1, P_2, \cdots P_v) = \sum_{L_v' m_3} w_3^{L_v' M_v} \mathrm{irreps}_3^{L_v' M_v}
$$

$$
= \sum_{L_v' M_v} w_3^{L_v' M_v} \sum_{L_{v-1}' M_v, \ell_v m_v} C_{L_{v-1}' M_v, \ell_v m_v}^{\ell_3 m_3} \mathrm{irreps}_1^{L_{v-1}' M_v} \mathrm{irreps}_2^{\ell_v m_v}
$$

$$
= \sum_{L_{v-1}' M_v \ell_v m_v} \mathrm{irreps}_1^{L_{v-1}' M_v} \mathrm{irreps}_2^{\ell_v m_v} \sum_{L_{v-1}' M_v} C_{L_{v-1}' M_v, \ell_v m_v}^{L_v' M_v} w_3^{L_v' M_v} \tag{26}
$$

Then we take $w_3^{L'_v M_v} = \sum_{\ell_1 m_1, \ell_2 m_2} C_{L'_{v-1} M_{v-1}, \ell_v m_v}^{L_v M_v} w_1^{L'_{v-1} M_{v-1}} w_2^{\ell_v m_v}$, and with similar procedure to Equation 24, we can derive that

$$
\begin{aligned}
&Q'^{(\mathbf{t_1}, \mathbf{t_2}, \cdots, \mathbf{t}_v)}(P_1, P_2, \cdots P_v) \\
&= \sum_{L'_{v-1} M_v \ell_v m_v} w_1^{L'_{v-1} M_{v-1}} \mathrm{irreps}_1^{L'_{v-1} M_v} w_2^{\ell_v m_v} \mathrm{irreps}_2^{\ell_v m_v} \\
&= Q^{(\mathbf{t_1}, \mathbf{t_2}, \cdots, \mathbf{t}_{v-1})}(P_1, P_2, \cdots P_{v-1}) Q^{\mathbf{t}_v}(P_v)
\end{aligned}
\tag{27}
$$

Above all, the output of equivariant base can represent $Q^{(\mathbf{t_1}, \mathbf{t_2}, \cdots, \mathbf{t}_v)}(\boldsymbol{\theta}_i)$ with $\|\mathbf{t}_j\|_1 \leq v, \forall j \in [v]$. Since it can select any $(\mathbf{t_1}, \mathbf{t_2}, \cdots, \mathbf{t}_v)$ within $\|(\mathbf{t_1}, \mathbf{t_2}, \cdots, \mathbf{t}_v)\|_1 \leq v$, the constructed D-spanning family $Q_K^D$ with $K = v$ and $D = v$. Then, the set of the output equivariant features is also a D-spanning family with $D = v$.

### A.2.4 PROOF OF D-SPANNING IRREPS FOR POLYNOMIAL MB INTERACTION MODULE

*Proof.* From proof A.2.2, each input $\mathbf{A}_i$ can represent $\mathbf{Q}^{\mathbf{t}}(\boldsymbol{\theta}_i)(P)$. Then when first channel irreps represent $\mathbf{Q}^{\mathbf{t}}(\boldsymbol{\theta}_i)(P_1)$, and second channel irreps represent $\mathbf{Q}^{\mathbf{t}}(\boldsymbol{\theta}_i)(P_2)$, with self-interaction layer before each atomic basis, $\mathbf{A}_{iv}$ can select different channels of the input $\mathbf{A}_i$. From Theorem 2, the output of polynomial MB module can represent $Q^{(\mathbf{t_1}, \mathbf{t_2}, \cdots, \mathbf{t}_v)}(\boldsymbol{\theta}_i)$ which then construct D-spanning family $Q_K^D$ with $D = K = v$. Therefore, the output irreps from the proposed polynomial MB interaction module can build the D-spanning family. $\qquad \square$

### A.2.5 MACE

For the MACE layer, it is built related to the channel coupling relationship discussed in the section of BOTNet (Batatia et al., 2022a), and tensor decomposition module discussed in (Darby et al., 2023). Next, we try to analyze the MACE layer considering the capacity of the output irreducible representation to span $D$-spanning family.

Consider the case of using tensor representations and input basis $A_i$ satisfies the condition in Theorem 1, denoted as $Q_K^{(\mathbf{t})}(\boldsymbol{\theta}_i)(P_1) = \sum_{\ell m} w_1^{\ell m} \mathrm{irreps}_1^{\ell m}$. Note that the $\mathrm{irreps}_1$ contains single channel for various rotation orders. We first apply similar analysis to the many-body interaction module in MACE named tensor decomposition schema (Darby et al., 2023). In this module, the same irreducible representations is applied when conducting tensor product over the path. In this module, the same irreps are used in many-body interactions. With similar proof shown in Lemma 1, the output feature in this case can represent $\mathbf{Q}^{(\mathbf{t_1}, \mathbf{t_1})}(\boldsymbol{\theta}_i)(P_1, P_1)$ instead of $\mathbf{Q}^{(\mathbf{t_1}, \mathbf{t_2})}(\boldsymbol{\theta}_i)(P_1, P_2)$. That's to say, MACE layer does not satisfy the necessary condition that the output irreps can represent $D$-spanning family $\mathbf{Q}^{(\mathbf{t_1}, \mathbf{t_2})}(\boldsymbol{\theta}_i)(P_1, P_2)$. As a result, we can not conclude that MACE layer can approximate the equivariant functions with proposed analysis tools.

Furthermore, we consider the ability of MACE layer for approximating the equivariant polynomial functions by approximate the output irreducible representations with an architecture that has already been proved to ensure the capability of approximating equivariant polynomial functions. As proved in Darby et al. (2023), the approximation error incurred in compressed least-square regression are expected to decay with $\frac{1}{\sqrt{K}}$, where $K$ is the number of channels. That's to say, the tensor decomposed product basis in MACE layer can always recover tensor sketched basis with infinite number of channels. Meanwhile, tensor sketched basis is the same operation used in the many-body module in PACE layer. In conclusion, with infinity number of channels, MACE layer can recover same ability compared to PACE layer to approximate the equivariant polynomial function $\mathcal{P}_G^D(\mathbf{C})$.

### A.2.6 NEQUIP

For the NequIP, the output irreducible representation from a single layer can represent $Q^{\mathbf{t_1}}(\boldsymbol{\theta}_i) = \sum_j \mathbf{r}_{ij}^{\otimes \mathbf{t_1}}$. Since $Q^{\mathbf{t_1}}(\boldsymbol{\theta}_i) = \sum_j \mathbf{r}_{ij}^{\otimes \mathbf{t_1}}$ can build the D-spanning family $D_K^D$ with $K = 1$. Therefore, with $D \leq K$, it can achieve $D = 1$ for a single layer.

### A.2.7 ALLEGRO

Similar to the NequIP, the edge equivariant representation $\text{irreps}_1$ can represent $Q^{\mathbf{t}_1}(\boldsymbol{\theta}_i) = \sum_j \mathbf{r}_{ij}^{\otimes \mathbf{t}_1}$. Then in the next layer, it can represent $Q^{(\mathbf{t}_1, \mathbf{t}_2)}(\boldsymbol{\theta}_i) = \sum_{j_1, j_2} \mathbf{r}_{ij_1}^{\otimes \mathbf{t}_1} \otimes \mathbf{r}_{ij_2}^{\otimes \mathbf{t}_2}$ with the Lemma 1. For the $m$-th layer, the input irreps can represent $Q^{(\mathbf{t}_1, \mathbf{t}_2, \cdots, \mathbf{t}_{m-1})}(\boldsymbol{\theta}_i)$, with Lemma 1, the output representation can represent $Q^{(\mathbf{t}_1, \mathbf{t}_2, \cdots, \mathbf{t}_m)}(\boldsymbol{\theta}_i)$. Therefore, $K = N_{\text{layer}}$ for D-spanning family $Q_K^D$.

## B  TENSOR CONTRACTION IN POLYNOMIAL MANY-BODY INTERACTION MODULE

In Figure 4, we illustrate tensor contraction which is originally proposed in (Batatia et al., 2022b) using an example, in which the max correlation order $v_{\max}$ is 3, the output $LM$ is $L_3M_3$, and the number of hidden channels is only 1 for easier illustration. Each generalized CG matrix corresponds to one path $\eta = (\ell_1, \ell_2, L_2, \cdots, \ell_v, L_v)$ with $L_1 = \ell_1$, and each path is associated with a scalar weight. Note that more hidden channels introduce a higher dimensional weight vector. A weight contraction followed by a feature contraction leads to efficient computation.

## C  EXPERIMENT DETAILS

Data and splits of rMD17 dataset are downloaded from `https://figshare.com/articles/dataset/Revised_MD17_dataset_rMD17_/12672038`. The unit of energy is converted from kcal/mol to meV. 3BPA dataset is downloaded from `https://github.com/davkovacs/BOTNet-datasets/tree/main/dataset_3BPA`.

For all molecules in rMD17 and 3BPA, we use 2 GNN layers with 256 hidden channels. The rotation order of node features is 2. Edges are built for pairwise distances within a radius cutoff. Edge features are initiated with either Bessel basis functions or Exponential Bernstein radial basis functions. SiLU is used as nonlinearity. We use a batch size of 5, an initial learning rate of 0.01, and Adam-AMSGrad optimizer with default paramters of $\beta_1 = 0.9$, $\beta_2 = 0.999$, $\epsilon = 10^{-8}$, and without weight decay. The learning rate is reduced on-plateau scheduler based on the validation loss with a patience of 100 and a decay factor of 0.8. We also use an exponential moving average with weight 0.99. The weight of force in loss is 1000 for all molecules, while the weight of energy varies depending on molecules. Table 4 shows our options for molecules in rMD17 dataset. For 3BPA dataset, edges are built using a radius cutoff of 5 Å, and edge features are initiated with 8 Bessel basis functions. The ratio of energy and force in loss is 15:1000.

Table 4: Model architectural hyperparameters for rMD17.

|  | Aspirin | Azobenzene | Benzene | Ethanol | Malonaldehyde | Naphthalene | Paracetamol | Salicyclic acid | Toluene | Uracil |
|---|---|---|---|---|---|---|---|---|---|---|
| Edge embedding | Bessel | Bessel | Bessel | EBRadial | EBRadial | Bessel | Bessel | Bessel | Bessel | Bessel |
| # of basis | 8 | 8 | 8 | 16 | 20 | 8 | 8 | 4 | 8 | 8 |
| Radius (Å) | 5 | 5 | 5 | 5 | 6 | 5 | 5 | 5 | 5 | 6 |
| Energy weight | 21 | 9 | 9 | 9 | 9 | 5 | 9 | 9 | 9 | 9 |

## D  RESULTS OF MORE EMPIRICAL ANALYSES

### D.1  THE ACAC DATASET

The AcAc dataset (Batatia et al., 2022a) is also generated through molecular dynamic simulation using Density Functional Theory (DFT). Similar to 3BPA, this dataset specifically focuses on a single molecule, Acetylacetone. We follow MACE to train our model using a training set comprising of 450 structures and a validation set comprising of 50 structures. Both the training and validation sets were sampled at 300K, while two test datasets were sampled at two different temperatures: 300K and 600K. The experimental results in Table 5 show that our proposed PACE achieves state-of-the-art results on this AcAc dataset.

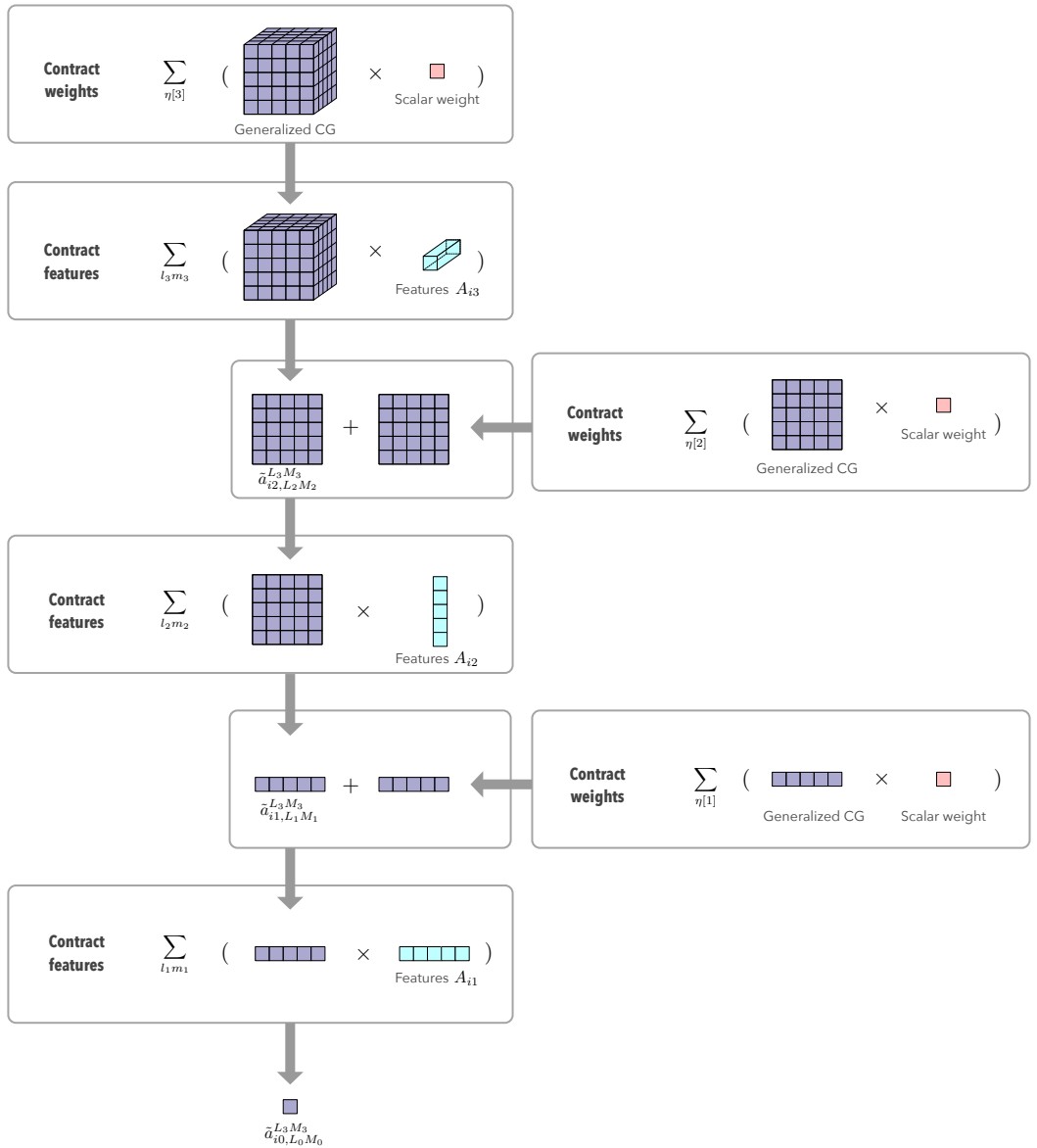

Figure 4: Illustration of tensor contraction in the polynomial many-body interaction module. In this figure, it demonstrates an example of 4-body interactions with $v = 3$ and final $L_3 = 0, M_3 = 0$. Note that the contract weights operation learns weighted summation over all paths $\eta[v]$, where $\eta[v] = (\ell_1, \ell_2, L_2, \cdots, \ell_v, L_v)$.

## D.2 MOLECULAR DYNAMIC SIMULATION

To further assess the ability of PACE to simulate realistic structures and dynamics, we conduct Molecular Dynamics (MD) simulations using PACE. These simulations are implemented with the Langevin dynamics provided by the ASE library and are applied to three molecules: Aspirin, Ethanol, and 3BPA. We randomly select an initial structure from the testing set to generate MD trajectories with PACE. Correspondingly, we extract the ground truth MD trajectory, starting from the same initial structure, from the testing set. The radial distribution functions (RDFs) of the ground truth MD trajectories and PACE-generated MD trajectories are visualized in Figure 5. Additionally, to compare the MD trajectories generated by PACE with those from our baseline MACE, we compute Wright's

Table 5: Performance on the AcAc dataset. Root-mean-square errors (RMSE) are reported for both energy (E) and force (F) predictions, with meV and meV/Å as units, respectively. Standard deviations are calculated over three runs with different seeds. Bold numbers highlight the best performance.

|      |   | BOTNet     | NequIP      | MACE       | Ours       |
|------|---|------------|-------------|------------|------------|
| 300K | E | 0.89 (0.0) | **0.81** (0.04) | 0.9 (0.03) | **0.81** (0.05) |
|      | F | 6.3 (0.0)  | 5.90 (0.38) | 5.1 (0.1)  | **4.8** (0.3) |
| 600K | E | 6.2 (1.1)  | 6.04 (1.26) | 4.6 (0.3)  | **4.3** (0.3) |
|      | F | 29.8 (1.0) | 27.8 (3.29) | 22.4 (0.9) | **21.2** (0.7) |

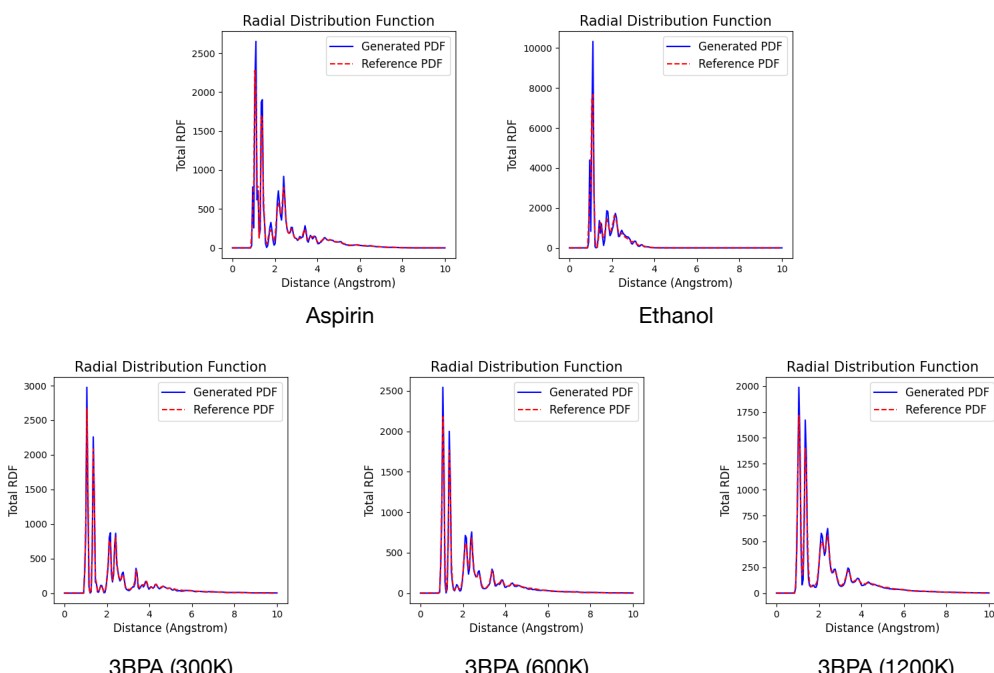

Figure 5: Illustration of radial distribution functions of MD trajectories. Values are averaged over five MD simulations for 1000 timesteps with five initial molecular structures.

Factor (WF) (Grimley et al., 1990), which is defined as

$$R_\chi = \frac{\sum_i^n (T_g(r_i) - T_{ref}(r_i))^2}{\sum_i^n (T_{ref}(r_i))^2}, \qquad (28)$$

where $T$ denotes the radial distribution function and $n$ denotes the total number of bins. Results presented in Table 6 are calculated using distance $r$ ranging from 0 to 10 Å and bin size of 0.05 Å. Here, WF quantitatively measures the difference between the RDFs of the ground truth and the generated MD trajectories. The results of the MD simulations further affirm that our PACE model not only achieves state-of-the-art (SOTA) performance in force field prediction but also holds practical value in realistic simulations.

## D.3 ABLATION STUDY

The first ablation study is for demonstrating the effectiveness of additional self-interaction operations introduced in our polynomial many-body interaction module. In this experiment, we removed additional self-interaction operations to map different atomic bases and used the same hyperparameters for model and training. The comparison shown in Table 7 justifies the effectiveness of the polynomial many-body interaction module in PACE.

Table 6: Performance of MD simulation. Wright's Factor (WF) is reported in %. Values are averaged over five MD simulations for 1000 timesteps with five initial molecular structures. Bold numbers highlight the best performance.

|  | MACE | Ours |
|---|---|---|
| Aspirin | 7.15 | **6.88** |
| Ethanol | 19.16 | **18.86** |
| 3BPA 300K | 2.14 | **1.98** |
| 3BPA 600K | 3.01 | **3.00** |
| 3BPA 1200K | 2.68 | **2.40** |

Table 7: The ablation experiments for self-interaction introduced in the polynomial many-body interaction module. Compared to PACE, PACE (-SI) uses no additional self-interaction. Note that both experiments share the same hyperparameters. Mean absolute errors (MAE) are reported for both energy (E) and force (F) predictions, with meV and meV/ Å as units, respectively. Bold numbers highlight the best performance.

|  |  | PACE (-SI) | PACE |
|---|---|---|---|
| Ethanol | E | 0.4 | **0.3** |
|  | F | 2.3 | **2.0** |
| Toluene | E | **0.2** | **0.2** |
|  | F | 1.4 | **1.2** |
| Uracil | E | **0.3** | **0.3** |
|  | F | 2.3 | **2.0** |

The second ablation study is for examing how degree $v$ in our polynomial many-body interaction module affects the prediction error and cost. In this experiment, we use the Ethanol dataset, and we use the same hyperparameters except the degree $v$ for fair comparison. Table 8 shows that higher degree archives lower prediction errors while consuming longer training time and more memory. Due to the tradeoff between error and cost, $v = 3$ is used in our PACE across all datasets.

## D.4 ALGORITHM EFFICIENCY

Table 9 presents a comparative analysis of training time and memory consumption between our proposed PACE method and the baseline methods. Results are reported in seconds per epoch and MB as units. A consistent batch size of 5 is used for each method, with average times calculated over 10 epochs, where validation occurs once every 2 epochs. In these experiments, Allegro is configured with 5 layers and a rotation order of 3, and NequIP with 3 layers and a rotation order of 3. Both MACE and PACE are configured with 2 layers and a rotation order of 2. The results indicate that the proposed PACE achieves universality and superior performance with advantageous computational cost.

Table 8: The ablation experiments for degree $v$ in the polynomial many-body interaction module. Mean absolute errors (MAE) are reported for both energy (E) and force (F) predictions, with meV and meV/ Å as units, respectively. The units for training time and memory consumption are sec/epoch and MD, respectively. Bold numbers highlight the best performance.

|  | $v = 1$ | $v = 2$ | $v = 3$ | $v = 4$ |
|---|---|---|---|---|
| E | 1.1 | **0.3** | **0.3** | - |
| F | 6.5 | 2.3 | **2.0** | - |
| Training time | 21 | 25 | 27 | - |
| Memory | 1583 | 1653 | 2283 | OOM |

Table 9: Training time and memory consumption, with sec/epoch and MB as units, respectively.

|  |  | NequIP | Allegro | MACE | Ours |
|---|---|---|---|---|---|
| Paracetamol | Training time | 130 | 50.3 | 28 | 34.5 |
|  | Memory | 3511 | 10211 | 4334 | 5080 |
| Toluene | Training time | 110 | 35.8 | 22 | 25 |
|  | Memory | 3507 | 3595 | 3386 | 3838 |

