# OpenReview forum: "Equivariant Graph Network Approximations of High-Degree Polynomials for Force Field Prediction"
_ICLR.cc/2024/Conference — Submitted to ICLR 2024_

### Official Review · Reviewer_8CoZ · 2023-10-29

**Soundness:** 2 fair
**Presentation:** 3 good
**Contribution:** 3 good
**Rating:** 6
**Confidence:** 3

**Summary:**

The paper proposes a new equivariant network which use the Atomic Cluster Expansion mechanism to encode many-body interactions. The proposed framework can provably approximate any equivariant polynomial functions with constrained degree. Experiments on several molecule benchmarks show the effectiveness of the model.

**Strengths:**

The paper is generally well-written and easy to follow.

The proposed model is well motivated, and surpasses previous models in terms of the ability to provably approximate equivariant high-degree polynomials.

The theoretical studies on approximating high-degree polynomial functions are appreciated.

**Weaknesses:**

1. I recommend adding some experiments to empirically show (1) whether PACE empirically approximates the equivariant high-degree polynomials; and (2) whether PACE can surpass existing works on approximating these polynomials.

2. Some other highly-related popular benchmarks, such as AcAc, which is adopted in (Batatia et al. 2022a; Batatia et al.2022b) should also be included.

3. I think Section 4.3 should be “Algorithm efficiency” instead of “Ablation study”.

**Questions:**

See Weakness.

---

> ### Author Response · Authors · 2023-11-20
> **Response to Reviewer 8CoZ**
>
> Dear reviewer 8CoZ, thank you for your valuable comments and positive review! Below are our responses to your concerns and questions.
>
> **W1.** Thanks for your comments. We will further consider this experiment in future work.
>
> **W2.** Thanks for your suggestion. During the discussion period, we conducted experiments on AcAc and added them to Appendix D1 in the revised paper.
>
> |      |   | BOTNet     | NeuqIP      | MACE       | Ours            |
> |------|---|------------|-------------|------------|-----------------|
> | 300K | E | 0.89 (0.0) | 0.81 (0.04) | 0.9 (0.03) | **0.81** (0.05) |
> |      | F | 6.3 (0.0)  | 5.90 (0.38) | 5.1 (0.1)  | **4.8** (0.3)   |
> | 600K | E | 6.2 (1.1)  | 6.04 (1.26) | 4.6 (0.3)  | **4.3** (0.3)   |
> |      | F | 29.8 (1.0) | 27.8 (3.29) | 22.4 (0.9) | **21.2** (0.7)  |
>
> **W3.** Thank you for your advice. Since we performed more experiments this week, we modified Section 4.3 to "More Empirical Analyses", and now "Algorithm Efficiency" is under it. The text and table of "Algorithm Efficiency" are moved to Appendix D4 in the revised paper.
>
> We appreciate it very much for your time and efforts to evaluate our paper! Hope our responses have properly addressed your concerns.
>
> Best regards,
>
> Authors

---

> > ### Comment · Reviewer_8CoZ · 2023-11-22
> >
> > Thanks for your detailed response, which helps me understand the paper better.

---

### Official Review · Reviewer_9963 · 2023-10-31

**Soundness:** 4 excellent
**Presentation:** 3 good
**Contribution:** 3 good
**Rating:** 8
**Confidence:** 4

**Summary:**

This paper introduces equivariant polynomial interactions, expanding the atomic cluster expansion framework and leading to improved performance on molecular dynamics datasets (rMD17). The work is well motivated and benchmarked.

**Strengths:**

The benchmark results are strong, method are well motivated, and the majority of paper is well written (improvements to make mentioned below).

**Weaknesses:**

The primary weakness -- which can easily be improved in the revision, is helping the reader understand and keep track of particular indices.

For section 2.3, I know that this is rephrasing of the ACE paper, but since later in the paper you treat the A's as equivariant and in the irrep basis to be contracted with CG coefficients, you may want to say something more about the equivariant nature of the c's and A's in this section to prep the reader and include a bit more foreshadowing about what elements are similar / different. I would also more clearly define the v indices as they are used throughout the paper.

Figure 3 is extremely helpful for understanding the equation below equation 10 -- I was unable to understand the equation without it. I know it takes up a huge amount of space, but even the second order version of the diagram would really help the reader parse the equation. I would prioritize Figure 3 over Figure 2.

**Questions:**

I do not have any questions.

---

> ### Author Response · Authors · 2023-11-20
> **Response to Reviewer 9963**
>
> Dear reviewer 9963, thank you for your valuable comments and positive review! Below are our responses to your concerns.
>
> **W1.** Thanks for your opinion. We have modified the corresponding part in the Section 2.3 denoted in red.
>
> **W2.** We have added the figure of tensor contraction above Equation 11. Thanks to your advice, here we made a 3-body interaction example to save space.
>
> Please see the above modifications in the revised paper. We appreciate it very much for your time and efforts to evaluate our paper! Hope our responses have properly addressed your concerns.
>
> Best regards,
>
> Authors

---

### Official Review · Reviewer_Nhdf · 2023-10-31

**Soundness:** 2 fair
**Presentation:** 2 fair
**Contribution:** 2 fair
**Rating:** 6
**Confidence:** 4

**Summary:**

The paper proposes a new equivariant network called PACE to incorporate many-body interactions by making use of the Atomic Cluster Expansion (ACE) mechanism. The authors provide a theoretical analysis, demonstrating that the proposed message-passing scheme can approximate any equivariant polynomial functions with a constrained degree. PACE can achieve state-of-the-art performance on several atomic potential and force field prediction tasks.

**Strengths:**

* The proposed message-passing scheme, which effectively approximates equivariant polynomial functions within a constrained degree, is a noteworthy contribution. The accompanying theoretical analysis is well-grounded.
* The paper is well-organized and easy to follow.

**Weaknesses:**

* The empirical analysis could be enhanced to better support the motivation. As I understand, the primary contribution of PACE, as compared to MACE, is the introduction of self-interaction layers that map the atomic base $A_i$ to different subspaces for different body orders, thereby increasing expressive power of PACE. However, the paper lacks an ablation study directly analyzing the significance of this operation. Although MACE could be considered a natural ablative baseline, it's unclear whether the two models share the same hyper-parameters, such as those mentioned in Table 5. I recommend that the authors conduct a direct ablation experiment to clarify this point.

**Questions:**

* Here is a sentence not easy to follow. In the 3rd paragraph in sec 3.1, ``While TFN has the ability to approximate all equivariant polynomial functions with an infinite number of layers, the approximation capacity of the equivariant message passing scheme is inherently limited.'' The authors should clarify the logic between these sentences.
* Considering that all baseline models except MACE possess the universal approximation ability for all equivariant polynomials, it's puzzling why they do not achieve performance comparable to PACE. Could the authors elaborate on this?

---

> ### Author Response · Authors · 2023-11-20
> **Response to Reviewer Nhdf**
>
> Dear reviewer Nhdf, thank you for your valuable comments and positive review! Below are our responses to your concerns and questions.
>
> **W1.** Thank you for suggesting the direct ablation experiment. During the discussion period, we conducted this ablation experiment on three molecules from rMD17, and the comparison is shown below. In this experiment, we removed additional self-interaction operations and used the same hyperparameters. The results confirm the effectiveness of our polynomial many-body interaction module. We have added this ablation experiment to Appendix D3 in the revised paper.
>
> |         |   | PACE (-SI) | PACE |
> |---------|---|------------|------|
> | Ethanol | E | 0.4        | **0.3**  |
> |         | F | 2.3        | **2.0**  |
> | Toluene | E | **0.2**        | **0.2**  |
> |         | F | 1.4        | **1.2**  |
> | Uracil  | E | **0.3**        | **0.3**  |
> |         | F | 2.3        | **2.0**  |
>
> **Q1.** Thanks for pointing it out. In the revised paper, we have modified this sentence to the following expression.
> “While TFN has the ability to approximate all equivariant polynomial functions with an infinite number of layers, the approximation capacity of a single TFN layer is limited, raising the need to develop modules improving the approximation capacity.”
>
> **Q2.** Thanks for this question.  We wish to clarify that these architectures can achieve universal approximation to all equivariant polynomial functions up to the highest degree $D$, denoted as $\mathcal{P}^{D}$. As shown in Table $1$, a single PACE layer can achieve $D=v=3$ in the experiment setup involving two layers, whereas a single NequIP can only reach $D=1$, and Allegro, with its three layers, attains $D=3$.
>
> We appreciate it very much for your time and efforts to evaluate our paper! Hope our responses have properly addressed your concerns.
>
> Best regards,
>
> Authors

---

> ### Comment · Reviewer_Nhdf · 2023-11-23
> **Thanks for the response**
>
> I am satisfied with the response to W1 and Q1&2. As for W1, I recommend the authors to place the results of MACE in the table, and list results of more molecules in rMD17 to verify the effectiveness of the proposed self-interaction module.
>
> Accordingly, I will keep my rating. Thank you!

---

### Official Review · Reviewer_LgUR · 2023-11-01

**Soundness:** 3 good
**Presentation:** 3 good
**Contribution:** 1 poor
**Rating:** 3
**Confidence:** 3

**Summary:**

This paper proposed an improvement over the Atomic Cluster Expansion (ACE) method, by introducing a polynomial many-body interaction layer in the neural network force field. It was theoretically proved that the polynomial many-body interaction layer can approximate any equivariant polynomial function with a fixed degree. Experiments were conducted on rMD17 and 3BPA datasets where the proposed method show a marginal improvement over the baseline methods.

**Strengths:**

The paper is written in a clear form and easy to read. The theoretical studies in Section 3.3 is solid.

**Weaknesses:**

-  The improvement in the experiments is so marginal. Recent years, people [1] have already noticed that achieving a small energy and force error is only the first step towards a successful neural network force field. The advantage of a neural network force field should be judged on the prediction of macroscopic properties and sampling of conformations on a long-time large-scale molecular dynamic simulation. An improvement on the energy/force error does not necessarily lead to a better MD prediction, unless the improvement is very obvious. Based on the numbers shown in Table 2 and 3, as well as the limit of the datasets, I'm not convinced that the proposed model is a better model than the baselines. Also the proposed model is slower than MACE, making it more useless in applications.

- The theoretical statement in this paper is not very important. Theorem 2 showed that the proposed polynomial many-body interaction module can approximate any polynomial equivariant functions. However, the goal of designing a neural network force field is to develop an end-to-end approximation to potential energy -- which is a non-polynomial equivariant function of the input coordinates. It is already proved that multi-layer perceptrons can already approximate any (polynomial or not) functions by the universal approximation theorem. So, no matter whether the polynomial many-body interaction module can approximate polynomial equivariant functions or not, the entire neural network is a universal approximator for any equivariant functions. This is why I think the value of Theorem 2 is limited.

**Questions:**

A key factor of the proposed neural network is the degree in the polynomial many-body interaction module denoted by v. I wonder how does the degree affects the error and the running time/memory cost of the model?

---

> ### Author Response · Authors · 2023-11-20
> **Response to Reviewer LgUR**
>
> Dear reviewer LgUR, thank you for your valuable comments! Below are our responses to your concerns and questions.
>
> **W1.** During the discussion period, we performed MD simulations using our PACE model on three molecules, Aspirin, Ethanol, and 3BPA. To compare the MD trajectories generated by PACE with those generated by our baseline MACE, we compute Wright's Factor (WF) [1] as a metric and present the comparison in the table below. Here, WF quantitatively measures the difference between the RDFs of the ground truth and the generated MD trajectories. For each method, we take the average of 5 MD simulations for 1000 timesteps with 5 different initial structures.
>
> |            |  MACE |      PACE |
> |------------|------:|----------:|
> | Aspirin    |  7.15 |  **6.88** |
> | Ethanol    | 19.16 | **16.86** |
> | 3BPA 300k  |  2.14 |  **1.98** |
> | 3BPA 600k  |  3.01 |  **3.00** |
> | 3BPA 1200k |  2.68 |  **2.40** |
>
> In addition, we also visualize the radial distribution functions (RDFs) of the ground truth MD trajectories and PACE-generated MD trajectories, which can be found in Figure 5 in the revised paper. These results of the MD simulations further affirm that our PACE model not only achieves state-of-the-art (SOTA) performance in force field prediction but also holds practical value in realistic simulations.
>
> We have added MD experiments to Appendix D2 in the revised paper. We agree that low energy and force errors do not guarantee better MD simulations. However, there’s still a lack of standards to evaluate the complex MD simulations. Currently, energy and force errors are still the most common metrics to assess the ability of machine learning models to predict force fields [2].  We will consider developing machine learning models more specifically for realistic MD simulations in future works.
>
> **W2.** We wish to clarify that our proposed theoretical statement aims to provide a tool to analyze the equivariant networks from the perspective of approximating equivariant polynomial functions. There are three points we wish to emphasize.
>
> 1. We wish to clarify that MLP is not an equivariant architecture. That’s to say, a simple MLP doesn’t satisfy the permutation, rotation and translation invariance or equivariance. Meanwhile, current 3D GNN architectures have shown these intrinsic symmetries are important when designing powerful practical model architectures. Therefore, the theoretical tool to analyze these invariant and equivariant architectures is important in our opinion.
>
> 2. Second, we wish to clarify that designing equivariant layer with good ability to approximate the equviariant polynomial function is not an easy task. Although the reviewer mentions that “no matter whether the polynomial many-body interaction module can approximate polynomial equivariant functions or not, the entire neural network is a universal approximator for any equivariant functions”, we hold a different opinion. Without the theoretical guidance for constructing the polynomial many-body interaction module, designing an equivariant layer with guaranteed and high capability to approximate the equivariant polynomial functions is challenging.
>
> 3. Finally, we wish to clarify that our proposed polynomial many-body interaction module in PACE can approximate any equivariant polynomial function with the highest degree $D=v=3$ with a single layer. Although the PACE layer needs more computational cost about 123% in training time and 127% in memory shown in Table $9$, we believe it is reasonable cost with carefully designed module under the guidance of obtaining the approximation capacity. Meanwhile, the final experimental performance is better.
>
> **Q1.** We performed an experiment on Ethanol to show how degree $v$ affects the error and cost. In this experiment, hyperparameters are the same. Results in the table below show that higher degree archives lower prediction errors while consuming longer training time and more memory. Due to the tradeoff between error and cost, $v=3$ is used in our PACE across all datasets. We have added this experiment to Appendix D3 in the revised paper.
>
> | Ethanol       |  v=1 |  v=2 |  v=3 | v=4 |
> |---------------|-----:|-----:|-----:|----:|
> | E             |  1.1 |  **0.3** |  **0.3** |   - |
> | F             |  6.5 |  2.3 |  **2.0** |   - |
> | Training time |   21 |   25 |   27 |   - |
> | Memory        | 1583 | 1653 | 2283 | OOM |
>
> [1]. Grimley, David I., Adrian C. Wright, and Roger N. Sinclair. "Neutron scattering from vitreous silica IV. Time-of-flight diffraction." Journal of Non-Crystalline Solids 119.1 (1990): 49-64.
>
> [2]. Hollingsworth, Scott A., and Ron O. Dror. "Molecular dynamics simulation for all." Neuron 99.6 (2018): 1129-1143.
>
> We appreciate it very much for your time and efforts to evaluate our paper! Hope our responses have properly addressed your concerns.
> Best regards,
>
> Authors

---

> ### Author Response · Authors · 2023-11-23
> **A Gentle Reminder**
>
> Dear Reviewer LgUR,
>
> The discussion deadline is approaching in a few hours. We have submitted our detailed responses to your comments and would greatly appreciate it if you could review our response and revised paper at your earliest convenience. Your feedback is invaluable to us, and we are eager to address any further comments or concerns you may have before the discussion period ends. Thank you very much for your time and consideration!
>
> Best regards,
>
> Authors

---

### Public Comment · ~David_Peter_Kovacs1 · 2023-12-11
**Concerns regarding the manuscript**

Dear AC and Reviewers,

We are writing this public comment to provide some context and additional information alongside this submission.
Note that we got in touch with the authors via email, highlighting some of our concerns. As a result, they significantly updated the manuscript based on our discussions. This happened after the review process ended. Nonetheless, many of our concerns still remain and we advise that the manuscript would benefit from further review.

- The manuscript builds on our MACE layer, and the difference is the treatment of the (radial and element) embedding channels. However, the index corresponding to the channel is not displayed in any of the equations. Therefore, it is difficult to follow how the new proposed computation layer works due to undefined symbols and indices in the equations. There are several symbols that appear and disappear without any accompanying explanation in the text. Some concrete examples of undefined and missing symbols are:
    - Eq 8, \( b \) is undefined and there is no radial or channel index?
    - Eq 9, why did it change to \( b=3 \), still no channel index, why \( A_i \) is not bold (is it a scalar here?)
    - Eq 11, why \( A_i \) now has 3 more indices, including \( v \). There is no equation showing how the authors get from \( A_i \) in Eq. 9 to \( A_{iv} \) in Eq. 11.
    - Eq. 12, \( x^{1}_{i,00} \) is undefined. The paper does not state what is \( 00 \) corresponding. Before, \( x \) never had more than two indices, but here has four, with the last two at the bottom presumably corresponding to the angular channels (?).
    - Why in Eqs. 7-8, the superscript is the \( l \) (angular) index, and in Eq. 12 seems to be the layer index.
    - There is no mention of how the authors go from the message in Eq. 11 to (presumably) update the node features for Eq. 12
- In their original submission, the authors missed two crucial references that contain a mathematical analysis of tensor product-based architectures [2, 3]. In Ref[2], which is now cited in the updated manuscript, it is shown that all the methods in this family (ACE, MACE, Nequip, Allegro etc., including the present proposal) have exactly the same polynomial expressivity with a finite number of channels. Crucially, this means that the PACE layer has exactly the same expressivity as MACE. Therefore, the updated discussion on this in Appendix A2.5 of the manuscript is still incorrect. This was also analyzed and discussed in more generality in two previous publications [1, 3]. Some of the key theorems about symmetric tensor decompositions are found in Ref[4] with rigorous proofs showing that any finite symmetric tensor can be written in a tensor decomposed form exactly.
- Given that it is now acknowledged in the revised version of the manuscript that an analysis of various tensor decompositions has been published before, the main contribution of this submission is the benchmarking of one of these ideas. However, the authors claim that the proposed architecture has higher expressivity than other tensor product architectures.
- In the updated version, Table 1 MACE entry was removed. However, all tensor-reduced ACE layers, including the MACE and PACE layers, should have the same entry (as far as it is possible to infer given the uncertainties of the indexing and missing equations in the manuscript).

We thank the AC and reviewers for considering our concerns.

References:

1. The Design Space of E(3)-Equivariant Atom-Centered Interatomic Potentials, Batatia et al, [https://arxiv.org/abs/2205.06643](https://arxiv.org/abs/2205.06643)
2. Tensor-reduced atomic density representations, Darby et al, PRL 131, 028001 (2023), [https://doi.org/10.1103/PhysRevLett.131.028001](https://doi.org/10.1103/PhysRevLett.131.028001)
3. Equivariant Tensor Networks, Hodapp et al, [https://arxiv.org/abs/2304.08226](https://arxiv.org/abs/2304.08226)
4. Symmetric tensors and symmetric tensor rank, Comon, Pierre, et al. SIAM Journal on Matrix Analysis and Applications 30.3 (2008):

---

### Meta-Review · Area_Chair_9FVc · 2023-12-15

**Metareview:**

In this paper a new equivariant architecture, PACE, is proposed to incorporate many-body interactions with the Atomic Cluster Expansion (ACE) mechanism. The paper shows theoretically that the message passing of the algorithm can approximate any equivariant polynomial function. The reviews were mostly positive, except for one review that questioned the importance of the results given marginal improvements in macroscopic properties and sampling of conformations on a long-time large-scale molecular dynamic simulation, not the accuracy of predicting the dynamic field itself. Authors run more experiments showing that the proposed model in fact offers reasonable improvement in accuracy also in this case. Other reviewers also requested ablation studies and extra confirmation experiments, which all were positive.

However, at the same time there was a separate public comment from the authors of MACE casting doubts on the contribution of the paper, specifically whether PACE is just as expressive as MACE and not more. Also, there was a remark that MACE was removed from Table 1 after the original submission. Considering that there exist significant concerns regarding as strong a claim as the 'universality of the method', the paper requires at least one more round of careful reviewing.

**Justification For Why Not Higher Score:**

Unclear contributions.

**Justification For Why Not Lower Score:**

See above

---

### Decision · Program_Chairs · 2024-01-16

Reject